# Painful Diabetic Neuropathy: Sex-Specific Mechanisms and Differences from Animal Models to Clinical Outcomes

**DOI:** 10.3390/cells13232024

**Published:** 2024-12-07

**Authors:** Emma Merlin, Chiara Salio, Francesco Ferrini

**Affiliations:** 1Department of Veterinary Sciences, University of Turin, Largo Braccini 2, 10095 Grugliasco, TO, Italy; emma.merlin@unito.it (E.M.); chiara.salio@unito.it (C.S.); 2Department of Psychiatry and Neurosciences, Université Laval, Québec, QC G1K 7P4, Canada

**Keywords:** diabetes, sex differences, preclinical models, clinical studies, pain assessment, neuropathic pain

## Abstract

Diabetes is a chronic and progressive disease associated with high blood glucose levels. Several co-morbidities arise from diabetes, the most common and severe one is diabetic neuropathy whose symptoms also include pain hypersensitivity. Currently, there are no effective therapies to counteract painful diabetic neuropathy or slow down the progression of the disease, and the underlying mechanisms are yet to be fully understood. Emerging data in recent decades have provided compelling evidence that the molecular and cellular mechanisms underlying chronic pain are different across the sexes. Interestingly, relevant differences have also been observed in the course and clinical presentation of painful diabetic neuropathy in humans. Here, we reviewed the current state of the art on sex differences in diabetic neuropathy, from animal models to clinical data. Comparing the output of both preclinical and clinical studies is necessary for properly orienting future choices in pain research, refining animal models, and interpreting clinical data. The identification of sex-specific mechanisms may help to develop more targeted therapies to counteract pain symptoms in diabetes.

## 1. Introduction

Diabetes mellitus has been referred to as the largest global epidemic of the 21st century [1], affecting over half a billion people, a number that is expected to increase by about 50% by 2075 [2]. Diabetes is a chronic, metabolic disease caused by elevated blood glucose levels that can be associated with an impairment of insulin production (Type 1 diabetes or T1D) or insulin resistance (Type 2 Diabetes or T2D). T2D is the most common type, accounting for ~90% of all cases [3]; on the other hand, in 2021, around 8.5 million people were diagnosed with T1D, among which 1.5 million were under 20 [4].

Diabetic neuropathy (DN) is one of the most common and debilitating co-morbidities affecting diabetic patients [5]. The global incidence of DN varies, but it is estimated that about 50% of people with diabetes develop some form of neuropathy over their lifetime [6]. Specifically, about 30–50% of individuals with T1D may experience DN after 20 years of living with the disease [7], while it affects 20–30% of T2D patients shortly after diagnosis, with an increase to over 50% after 10 years [6].

DN symptomatology is mainly associated with microvascular and metabolic alterations in the peripheral nervous system (PNS) [8,9]. Indeed, high blood glucose levels play a direct role in nerve cell damage leading to peripheral nerve dysfunction and/or death [10]. Particularly in T1D, uncontrolled glucose levels strongly correlate with the incidence and severity of DN [11]. Nerve damage is also favored by microvascular alterations and impaired insulin signaling [6,12] but several other conditions may act as risk factors, such as dyslipidemia and its associated oxidative stress [13].

Symptoms of DN are manifestations of the dysfunction of the somatosensory and motor systems involved and include numbness, tingling, pain, weakness, as well as impaired balance and coordination. Clinical features vary according to the type and length of nerve fibers involved: the more common peripheral polyneuropathy, which affects long peripheral nerves, causes pain or a loss of sensation in the extremities of the limbs; the autonomic neuropathy affects the functions of internal organs, such as the bowel, bladder, lungs, and heart; the focal neuropathy affects single nerves in the body, causing localized pain or muscle weakness [6].

Although several patients are asymptomatic, DN can result in both a gain of function (hypersensitivity) or a loss of function (hyposensitivity) symptoms [14,15]. Alterations affecting small-sized thinly myelinated Aδ fibers and/or unmyelinated C fibers, which are responsible for temperature and nociceptive stimuli transduction, lead to hyperalgesia and allodynia (increased sensitivity to painful and innocuous stimuli, respectively). Conversely, dysfunctions of highly myelinated Aβ fibers, mainly transducing innocuous mechanosensation, result in sensory ataxia and decreased proprioception [16]. Thus, DN could be either painful or painless. Painful DN develops in approximately 30–50% of people with DN, particularly with T2D [17,18,19], and involves the onset of neuropathic pain symptoms, including burning, pins, and electric shock-like pain [20,21]. Painless DN, instead, is characterized by sensory loss, which can lead to life-threatening complications such as injury, infection, and ulcers [18]. Both forms can be observed in the same patient during the course of the pathology and early diagnosis is required to prevent complications.

The mechanisms leading to painful or painless DN are still poorly understood and many inflammatory markers show no significant variation between diabetic patients with and without pain symptoms, so a strong causal link drawing a connection between diabetes-induced chemical or physical alterations and the presence of pain is missing [22].

The involvement of the somatosensory system following DN has a particularly strong impact on mechanical sensitivity, affecting the encoding of tactile stimuli, vibration, and proprioception [5,23]. The impact of DN can be significant and multifaceted, and can affect several personal and social aspects such as quality of life, mental health [6], relationships, employment, economic well-being, and healthcare utilization [24].

Despite efforts to achieve early diagnosis and to counteract the progression of DN, there is currently no truly effective treatment. The therapies used to treat pain in diabetic patients are the same as for neuropathic pain, and being poorly targeted, their effectiveness is only partial [6,25].

## 2. Sex Differences in Chronic Pain

Until recently, most of the preclinical literature on pain mechanisms was strongly based on male-centered animal models. A survey on the scientific articles published in 2010 [26] showed that only one out of five studies in the field of neuroscience used female animals. The bias toward male animals was typically justified by the supposed higher variability of females due to cyclic hormonal fluctuations, in spite of the fact that different neurological diseases, including chronic pain, are more common in females [27,28]. In recent decades, the awareness that conclusions drawn in males might not apply to females has been increasingly supported by several key studies showing sex-specific mechanisms underlying chronic pain [29,30].

Sex differences in chronic pain development are not only directly linked to gonadal hormones but are also associated with several different molecular and cellular players. The seminal paper by Sorge et al. pointed out the critical role of the immune system in the mechanisms underlying neuropathic pain in males and females [31]. In particular, spinal microglia, the resident macrophages in the central nervous system (CNS), were found to be necessary for the onset of pain hypersensitivity in males [32], but dispensable in females [31,33,34]. Conversely, specific molecular dimorphisms in T cells, involving the peroxisome proliferator-activated receptors (PPARs), appeared to be relevant for the onset of allodynia in females rather than in males [31]. Sex-specific functions have also been described for several neurotrophic factors and neuropeptides [35]. The brain-derived neurotrophic factor (BDNF), which has been consistently associated with neuropathic pain in males [36], does not underlie pain hypersensitivity in female rodents and humans [31,37]. Contrariwise, certain neuropeptides, such as prolactin and the calcitonin gene-related peptide (CGRP), are mainly involved in different forms of chronic pain in females, including migraine [38,39,40,41,42]. Also, fast neurotransmitters have been found to display sexually dimorphic effects [35]. GABAergic tonic currents mediated by α5-GABA_A_ receptors contribute to pain hypersensitivity in neuropathic pain due to altered chloride homeostasis, thus causing a tonic state of excitability at spinal circuits [43]. Interestingly, α5-GABA_A_ receptors are specifically upregulated in sensory neurons of female mice with chronic pain and are responsible of sex-specific pro-nociceptive effects [44]. The dopamine-dependent modulation of spinal neurons in neuropathic pain models is also sex-dependent, as it relies on D5 receptors in male mice and on D1 receptors in female mice [45].

However, in addition to an increasing number of sex-specific mechanisms, a wealth of data also highlights the commonalities. For instance, both chloride regulation and NMDA receptor functions, which are dramatically altered in pain-transmitting neurons following nerve injury [46,47], are as relevant in males as in females [31,48].

In the subsequent paragraphs, we will analyze the current knowledge on sex differences affecting the somatosensory system in subjects with DN, with a special focus on pain symptoms (or the lack of them). Data from both preclinical and clinical studies will be discussed in order to address the following questions: How effective are predictive experimental models of diabetes in detecting sex differences? And, on the other hand, can clinical studies orient the choices on future experimental designs?

## 3. Sex Differences in Diabetic Neuropathy: Lessons from Preclinical Studies

A search on PubMed performed on August 2024 for “diabetic neuropathy” OR “diabetic neuropathic pain” AND “sex” OR “sex differences” OR “gender” OR “gender differences” yielded 498 articles of which 26 were preclinical studies (mouse or rat). After further refinement, with the exclusion of those articles considering males only, or males and females pooled together, or that were focused on diseases unrelated to the somatosensory system (e.g., retinopathy or nephropathy), we have limited our selection to the 11 studies listed in Table 1; of these, 8 studied rats and 4 studied mice (only 1 study used both [49]).

Most of the selected studies (7 out of 11, [50,51,52,53,54,55,56]) adopt a chemically induced model of T1D (for a detailed description of animal models of diabetes see [57,58,59]). Specifically, the model is based on streptozotocin (STZ) administration which affects insulin production by targeting pancreatic β-cells [60,61]. Females appear more resistant to STZ administration than males [61], likely due to the protective effect of estrogens [62,63,64]. This is also consistent with the experience in our laboratory with C57BL/6 mice, confirming that higher doses of STZ are required to induce robust hyperglycemia in females (200 vs. 150 mg/kg, intraperitoneally; unpublished data and [65]). These differences in drug sensitivity across sexes should be carefully considered when defining the experimental design and must be taken into account for a correct data interpretation. Indeed, STZ itself, and not only the alterations induced by diabetes, may differentially affect organs and systems in males and females. On the other hand, the course and severity of hyperglycemia following STZ, once established, are similar across sexes [55,66] or more dramatic in females [50], while weight loss, a canonical hallmark of T1D, has been found to be more dramatic in male rats [55]. Moreover, emerging evidence suggests that sex differences may also be associated with insulin sensitivity, with female rats being less sensitive to insulin than males [67].

In terms of sensory behavior, both mechanical and thermal thresholds have been evaluated and compared across sexes [49,50,53,54,66,68]. A greater reduction in mechanical sensitivity, i.e., mechanical allodynia, has been typically described in female rodents [50,66]. Conversely, thermal hyperalgesia has been reported in male mice [66] and in both sexes (including gonadectomized animals) in rats [53]. Alterations in nociceptive behavior may derive from functional impairment in peripheral nerve physiology. Indeed, several studies have investigated nerve conduction velocities (NCV) and skin innervation in male and female rodents [53,54,66,69]. Male mice displaying thermal hyperalgesia show a greater decrease in NCV, unlike females [66]. Conversely, no sex differences in NCV or skin innervation have been reported in rats [53,54]. 

Protocols for STZ administration may differ in terms of the number of injections, the site of injection, the toxin volume [61], the age of the animals, the experimental time points, and the species, making any standardization difficult. Even in the relatively small number of studies identified here, different approaches are employed, namely, single [50,51,52,53,54,55] vs. multiple injections [66], intraperitoneal [53,54,66] vs. intravenous injections [50,51,52,55], different time points [50,51,52,53,54,55], and different species [50,51,52,53,54,55], [66].

To overcome some of the technical issues associated with chemically induced diabetes, several T1D and T2D models are based on genetic mutations. The Akita Ins2 mouse is a model of T1D diabetes in which a mutation of the gene-encoding insulin led to the misfolding of the proinsulin-2 protein [70]. Interestingly, male Akita mice display more severe diabetic symptoms than females (higher glycemia, higher mechanical sensitivity) and a stronger response to analgesic treatment. Other genetically based models reproduce obese forms of T2D, like those due to leptin deficiency (BTBR ob/ob mice, [71]) or leptin receptor deficiency (db/db mice, [72]; ZDK rats, [73]), or non-obese T2D models, like the polygenic model provided by the Goto-Kakizaki rats [74]. Unlike T1D models, little or no sex differences have been highlighted in T2D animals [49,56,69]. Also, in this case, male rodents develop signs of peripheral neuropathy earlier than females [69], although no sex-specific sensory profiles have been identified so far [49].

The involvement of sex hormones in shaping diabetic neuropathy has been duly investigated by Pesaresi et al. [52,53,54,55]. The overall neuroactive steroid levels decrease in both sexes at the peripheral and central levels in those with diabetes [53,54]. Interestingly, signs of peripheral neuropathy (e.g., NCVs) can be partly reversed by ovariectomy, but not orchiectomy, unmasking the relevance of sex hormones in the pathogenesis of diabetic neuropathy in females, but not in males [53]. These data suggest that overlapping symptoms and alterations in peripheral nerves may depend on totally different hormone-dependent mechanisms, which in turn may translate to different sex-specific therapeutic options. Indeed, diabetic female rats are more responsive to steroid treatments (specifically with dehydroepiandrosterone) than males [54]. Conversely, the reduction in testosterone in diabetic males may affect mitochondrial activity with direct consequences on axonal transport and nerve function [55]. Very little is known, however, on sex-specific cellular and molecular mechanisms underlying DN. In the seminal paper by Joseph and Levine [50], it was proposed that pain hypersensitivity in female rats was driven in sensory neurons by the second messenger protein kinase Cδ, while in males this was driven by protein kinase Cε. In T2D, several efforts have been made to identify a causal link between altered insulin receptor signaling and neuropathic pain symptoms. In a recent study, the role of PPARγ, which enhances insulin signaling and glucose uptake, was investigated [49]. As previously shown in neuropathic animals [31], PPARγ displayed a greater efficacy in restoring diabetic-induced pain in female db/db mice rather than males [49]. Since PPARγ has been previously linked with female immune system activation, these findings open new avenues in the roles of non-neuronal cells, such as T cells [75], in shaping sensory profiles across sexes in diabetes.

The high heterogeneity of the animal models for diabetes represents a relevant limitation for comparing data across studies and driving meaningful conclusions. In this respect, the age at which diabetes is induced and the experimental endpoints are quite relevant. Unfortunately, in the analyzed studies, animal age, when provided, typically ranges from 6 weeks [49,66] to 2 months [50,52,53,54,55,69]; in one case, it was expressed in terms of animal weight [56]. The duration of diabetes is even more changeable, varying from days [50], to several weeks [51,55], to several months [49,52,53,54,66]. Another limiting factor for a correct interpretation of the results is the lack of information about sex differences in control or naïve animals. Among the reviewed articles, information according to sex in the controls was available in two papers only [52,66].

Despite the intrinsic variability of the adopted models and the lack of data, some general considerations can be drawn: (1) male and female rodents with DN display different genetic, hormonal, and molecular pathways, sometimes leading to a similar set of symptoms and alterations in the sensory system; (2) estrogens are consistently found to be protective at the onset of DN; (3) mechanical allodynia, a classical symptom of neuropathic pain, is more often described in females with T1D.

**Table 1 cells-13-02024-t001:** Sex differences in diabetic neuropathy: lessons from preclinical studies.

Animal Model	Diabetic Model	TD	Hypothesis	Behavioral and Functional Tests	Sex Differences	Ref.
Sprague Dawley rats	Single i.v. injection of STZ 50 mg/kg	1	Role of PKC isoforms in diabetes-induced neuropathic pain	Mechanical nociceptive threshold	Glucose increase ♀ > ♂ Mechanical allodynia ♀ > ♂ In ♀, PKCδ In ♂, PKCε	[50]
Sprague Dawley rats	Single i.v. injection of STZ 35 mg/kg	1	Diabetes alters aromatase levels in PNS and CNS	None	Aromatase decreases in both sexes at 4 weeks after STZ Aromatase increases at 12 weeks in ♀	[51]
Sprague Dawley rats	Single i.v. injection of STZ 65 mg/kg	1	Diabetes alters neuroactive steroid levels in PNS and CNS	None	In sciatic nerve: pregnenolone, testosterone, and derivatives ♀ < ♂ progesterone and derivatives ♂ < ♀ Different patterns in CNS	[52]
Sprague Dawley rats	Single i.p. injection of STZ 60 mg/kg	1	Diabetes alters neuroactive steroid levels in PNS and CNS	Thermal nociceptive threshold NCV (tail)	Thermal hyperalgesia in both sexes and gonadectomized rats NCV was decreased in both sexes, but not in ovariectomized rats. Diabetes increases testosterone and derivatives in ovariectomized rats	[53]
Sprague Dawley rats	Single i.p. injection of STZ 60 mg/kg	1	DHEA has sex-specific neuroprotective effects	Thermal nociceptive threshold NCV (tail)	DHEA restores: thermal sensitivity ♀ > ♂ NCV ♀ > ♂ IENF ♀ > ♂	[54]
C57B6/L mice	3-day i.p. injection of STZ 85, 70, and 55 mg/kg	1	Intranasal insulin in well-established chronic experimental diabetic polyneuropathy influences the development of accepted diabetic polyneuropathy indexes	MCV, SCV Thermal and mechanical nociceptive threshold Rotarod testing Hindpaw grip	No sex differences in glucose and weight Slowing of MCV and SCV ♂ > ♀ Sensory nerve action potentials ♂ > ♀ In ♀ @16 wk, mechanical sensitivity increases In ♂ @ 8 wk, thermal sensitivity impaired Intranasal insulin: In both sexes @ 8wk, improvement of deficits in MCV and SCV In ♀ @ 16 wk, improvement in grip strength and mechanical sensitivity	[66]
Ins2 Akita mice	Insulin mutation	1	Soluble epoxide hydrolase reverses DN	Mechanical and thermal nociceptive threshold CPP	Hyperglycemia onset and severity ♂ > ♀ Mechanical allodynia ♂ > ♀ No thermal sensitivity in either sex CPP response after drug ♂ > ♀	[68]
Sprague Dawley rats	Single i.v. injection of STZ 60 mg/kg	1	Sex dimorphism in axonal transport alterations of peripheral nerves in diabetic rats	None	Hyperglycemia in both sexes Weight loss ♂ > ♀ Decrease testosterone and derivatives ♂ > ♀ Altered content of axonal motor protein ♂ > ♀	[55]
Wistar Goto-Kakizaki rats	Insulin resistance	2	Sexually dimorphism in nerve repair in diabetic rats	None	No sex differences in glucose and weight Axonal outgrowth after transection ♂ > ♀ Activated Schwann cells ♂ > ♀	[56]
BTBR *ob/+ ob/ob* mice	Leptin deficiency mutation	2	Sex differences in *ob/ob* mice diabetic peripheral neuropathy	NCV (sural and ischiatic nerve)	No sex differences in glucose, weight, and NCV Diabetic peripheral neuropathy in both sexes, but ♂ earlier than ♀ IENF ♂ > ♀ Hypertriglyceridemia ♂ > ♀	[69]
ZDF Rats *db/db* mice	Insulin resistance	2	PPARγ agonist has a sexual dimorphic effect in diabetic models	Mechanical and thermal nociceptive threshold	No sex difference in the development of blood glucose levels, weight, heat hypersensitivity or mechanical hypersensitivity No sex difference in the effect of PPARγ agonist ZDF rats, but in ♀ *db/db* mice more effective	[49]

Abbreviations: TD, type of diabetes; i.v., intravenous injection; STZ, streptozotocin; PKC, protein kinase C; i.p., intraperitoneal injection; PNS, peripheral nervous system; CNS, central nervous system; NCV, nerve conduction velocities; DHEA, dehydroepiandrosterone; IENF, intraepidermal nerve fiber density; MCV, motor nerve conduction velocities; SCV, sensory nerve conduction velocities; wk, week; DN, diabetic neuropathy; CPP, conditioned place preference.

## 4. Lessons from Clinical Studies: Is Diabetic Neuropathy Different in Men and Women?

Our PubMed search for sex differences in DN (previously described in detail) returned 338 clinical studies and 33 reviews. We subsequently selected those studies in which the impact of DN on the somatosensory system was investigated and data from men and women were separately analyzed.

Table 2 summarizes the findings of 22 epidemiological studies analyzing sex as a risk factor for the development of DN. Most of the studies (19 out of 22) are cross-sectional and retrospective studies mostly focused on T2D (11 out of 19) or, when specified (4 out of 19), on both types 1 and 2. Interestingly, female sex was identified as a risk factor for DN in eight of these cross-sectional studies [20,76,77,78,79,80,81,82], male sex was a risk factor in five studies [83,84,85,86,87], while no sex differences were found in six of the studies [88,89,90,91,92,93]. It is, however, important to note that only in eight of these studies was the pain profile associated with DN overtly investigated and analyzed [20,76,77,78,80,81,84,89], either by specific questionnaires (e.g., Douleur Neuropathique-4 questionnaire) or by clinical assessment. In the other studies [79,83,85,86,87,88,91], the main factors investigated are related to the loss-of-function symptoms associated with diabetes, including decreased tactile sensitivity (10 g monofilament), vibration perception, a reduction in sensory-motor reflexes (e.g., ankle reflex) as prescribed by the different scoring system (e.g., Michigan Neuropathy Screening Instrument, MNSI).

A longitudinal study performed on T2D young patients and focusing on DN reported that the male sex is a risk factor for this complication of diabetes [94]. Interestingly, when pain symptoms are also considered, the development of painful DN is more commonly observed in women [20,76,77,78,80,81] with an estimated 50% higher risk to develop neuropathic pain [20,78]. This evidence has been recently confirmed and supported by longitudinal and prospective studies involving both T1D and T2D patients [95,96]. Elliot et al. in a large longitudinal study, performed on over 3000 T1D patients followed up for 7 years, confirmed that the vast majority of them (about 73%) that were developing painful DN (defined as pain symptoms in lower limbs) were females [95]. Similarly, Abraham et al. showed that neuropathic pain symptoms are more commonly observed in females with T2D than males [96]. Interestingly, in this study, peripheral nerve functions (measured as compound action potentials) and vibration sensitivity were better preserved in females, likely as a consequence of female hormones’ protective effects [96]. On the other hand, men with T2D seem to develop pain symptoms a few years earlier than women [87]. These divergent data confirm that, when different aspects of the pathology are considered, different nuances in sex-dependent effects can also be detected. The increased incidence of pain symptoms in diabetic females, together with a more extensive loss of tactile and motor function in males, may reflect differential protective sex hormone effects in different components of the somatosensory system.

The methodology used for sensory and clinical profile assessment plays a pivotal role in shaping the outcome of clinical studies and the associated data interpretation. Most of the common methods for assessing DN do not properly probe pain symptoms. NCV is classically considered the gold standard in DN diagnosis, providing a direct readout of nerve function. However, baseline differences in conduction velocities and latencies have been described across sexes [97]. Moreover, it should be noted that NCV mainly assesses large fibers’ functioning [98], lacking specificity for small fibers activity and nociceptive thresholds. The widely used 10 g monofilament, used to assess the mechanical detection threshold when applied to the patient’s feet, has displayed a good capacity in predicting the incidence of DN, particularly in men [99]; on the other hand, the test has been shown to have low sensitivity in detecting neuropathy signs in women and in subjects with painful DN [100].

Similarly, most of the scoring systems for DN do not adequately account for nociception and pain perception. The use of a quantitative sensory test (QST), introduced by the German Research Network on Neuropathic Pain as a protocol for examining thermal and mechanical sensitivity [101,102], is more appropriate for exploring small fiber alterations and gain-of-function symptoms. Yet, its clinical use is still limited [95], mainly because it is time consuming and requires specialized equipment. 

Sex differences may also be relevant in shaping the course of co-morbidities associated with DN. In general, women exhibit higher levels of glycated hemoglobin (HbA1c), an indication of poor glycemic control [103]. Female sex, together with age, are the strongest predictive factors for the development of several co-morbidities in T1D, including thyroid gland disorders, urethritis, iron deficiency anemia, retinopathy, and ketoacidosis [104,105]. Females with late-onset T1D are also found to be more exposed to hospitalization for hypoglycemia than men [104]. Several co-morbidities affecting the motor system are associated with peripheral neurovascular alterations, particularly at limb extremities. Reduced muscle strength in lower limbs has been described in both males and females with DN [106], while reduced hand mobility, the development of foot ulcers, and the occurrence of foot amputation are more frequent in males [107,108,109]. On the other hand, testosterone is protective for the development of insulin resistance in male T1D patients, thus mitigating subsequent vascular complications [110]. 

Importantly, diabetes, DN, and diabetes-associated co-morbidities greatly affect the daily life of patients. Female DN patients reported a worse quality of life index than male patients [108] and a higher level of distress [111], which in turn may increase the risk of diabetic complications and poor glycemic control [112]. Moreover, women with DN are more exposed to developing cognitive impairment and psychiatric disorders than men [105,113,114,115].

Of note, none of the studies summarized in this paragraph take into account medical history and/or current hormonal conditions such as pregnancy, menstrual cycle, menopause, clinical hormone treatments, and contraceptive use in diabetic patients, although all these factors may have a relevant impact on the somatosensory response in women [116].

**Table 2 cells-13-02024-t002:** Sex as a risk factor in DN.

Study	TD	Cohort Size	Assessment	Pain Evaluation	Sex as a Risk Factor	Ref.
Longitudinal study (EURDIAB)	1	3250	Clinical assessment QST Autonomic function tests	Yes	♀ for painful DN (73%)	[95]
Longitudinal study adolescents (USA)	2	674	MNSI 10 g monofilament	No	♂ for DN (developing in 37% ♂ vs. 28% ♀)	[94]
Retrospective and prospective (Canada)	2	351	NCV Vibration perception Compound action potentials Cooling detection threshold	Yes	♀ for painful DN (68% ♀ vs. 53% ♂) ♀ > neuropathic pain in limbs ♀ > amplitude and conduction velocity	[96]
Cross-sectional (Saudi Arabia)	2	342	DN-4 questionnaire	Yes	♀ for painful DN (52%)	[76]
Cross-sectional retrospective (Saudi Arabia)	2	430	NDS DNS questionnaire Electromyography/NCV	No	♂ for DN	[83]
Cross-sectional (Japan)	2	9914	Diabetic Neuropathy Study Group in Japan criteria	Yes	♀ for painful DN (>50% risk)	[78]
Cross-sectional (India)	2	273	DNS questionnaire	Yes	♂ for DN ♂ (43%) > ♀ (27%) affected by painful DN	[84]
Cross-sectional (India)	2	586	10 g monofilament Pinprick sensations Ankle reflexes	No	No sex differences in DN	[88]
Cross-sectional observational (Korea)	2	1338	VAS Medical history BPI—SF MOS-Sleep Scale EuroQol (EQ-5D)	Yes	♀ for painful DN (prevalence: 46% ♀ vs. 39% ♂)	[77]
Cross-sectional (Sri Lanka)	2	528	DNS TCSS 10 g monofilament Pinprick sensations	Partly	♀ for DN (prevalence 26% ♀ vs. 20% ♂)	[79]
Cross-sectional Retrospective (Bulgaria)	2	1705	Medical history Overt neuropathic symptoms, Neurological tests	No	No sex difference in DN No sex differences in glycemic control ♀ for obesity, hypertension, and dyslipidemia	[91]
Cross-sectional (Iran)	2	110	MNSI NCV	No	♂ for DN	[85]
Retrospective analysis (UK, Germany)	2	14,000 UK 45,000 Germany	ICD	No	♂ (UK) for DN	[86]
Retrospective (USA)	2	376	ICD (9th revision), code: 250.60, 250.62	No	♂ develop DN earlier than ♀	[87]
Cross-sectional study (UK)	1,2	15,692	DNS NDS	Yes	♀ for painful DN (>50% higher risk)	[20]
Cross-sectional observational (Germany, Czech Republic)	1,2	232	Detailed history Laboratory tests Neurological examination QST NCV Neuropathy severity score	Yes	♀ for painful DN	[80]
Cross-sectional (Morocco)	1,2	300	DN-4 questionnaire	Yes	♀ for painful DN (♀/♂ ratio 2:1)	[81]
Cross-sectional (Bahrain)	1,2	1477	DNS NDS Questionnaire	Partly	No sex difference in DN (32% ♂ vs. 38% ♀)	[92]
Cross-sectional (Portugal)	ND	359	DN-4 questionnaire	Yes	No sex difference in neuropathic pain	[89]
Retrospective (Korea)	ND	40–60,000 per year	ICD (9th revision), code: G590 for diabetic mononeuropathy, G632 for diabetic polyneuropathy	No	No sex difference in DN	[90]
Cross-sectional (Canada; First Nation population)	ND	483	Laboratory tests Clinical examinations 10 g monofilament	No	♀ associated with DN	[82]
Cross-sectional (Nigeria)	ND	120	MNSI Vibratory sensation Ankle reflexes 10 g monofilament	Partly	No sex difference in DN	[93]

Abbreviations: QST, Quantitative Sensory Test; MNSI, Michigan Neuropathy Screening Instrument; DN-4, Douleur Neuropathique-4; NCV, Nerve Conduction Velocities; NDS, Neuropathy Disability Score; DNS, Diabetic neuropathy symptom; VAS, Visual Analogue Scales; BPI—SF, Brief Pain Inventory—Short Form, MOS, Medical Outcomes Study; ICD, International Classification of Disease; TCSS, Toronto Clinical Scoring System.

## 5. Preclinical to Clinical Studies in Diabetic Neuropathy and Reverse Translation

Translational research traditionally follows a "bench-to-bedside" approach where findings from preclinical studies—in vitro and/or in vivo—are developed and refined before being tested in clinical trials. Sometimes, the pathway switches in what is known as reverse translation or “bedside-to-bench” research. This approach starts with clinical observations and brings these insights back to the laboratory to explore underlying mechanisms. Reverse translation is particularly valuable in complex diseases, such as DN, where animal models need to be shaped around the variability found in the human population. To date, specific treatments for DN are still missing. At the same time, several promising molecular targets derived from preclinical studies failed clinical trials [117]. These failures prompted the re-examination of both animal models and clinical trial design [118].

Data from clinical studies have identified sex as an important risk factor in DN in humans. Research shows that the male sex may be associated with a higher risk of developing severe DN earlier in life [87]. Estrogens likely play a protective role in the onset of diabetes and its complications [119]. Indeed, premenopausal women display a lower prevalence of T2D [120]. On the other hand, men are less likely to develop neuropathic pain symptoms than diabetic women, which supports the anti-allodynic role of testosterone and its metabolites [121].

To fully understand the mechanisms underlying sex differences and to develop tailored treatments, it is strategic to adopt representative preclinical models. Unfortunately, the available data from diabetic animals display several inconsistencies and the role of sex as a risk factor does not appear as straightforward as in humans (Figure 1). Several lines of evidence support the protective role of estrogens in the onset of diabetes in different rodent models of T1D and T2D [62,122,123,124,125]. However, when changes in peripheral nerves and the somatosensory system are investigated, only a few of the available studies highlight a higher level of pain sensitivity in females [50,66], while others claim that there is more sensitivity in males or that there are no sex differences at all (Figure 1).

The inherent variability of the animal models and the still limited number of preclinical studies comparing sexes may be a reasonable cause that affects the robustness of current available data in the literature. More efforts should be made to improve and refine the analysis of sensory alteration induced by DN across sexes both at the laboratory bench and the bedside level.

In particular, major limitations of preclinical studies include the following:(1)The extensive use of chemically induced diabetic models typically leads to insulin depletion, thus representing a relevant bias toward T1D models. Although STZ-induced diabetes is a powerful and practical tool to achieve a high level of reproducibility and standardization, it may narrow the range of possible alterations related to diabetes [126]. Therefore, comparing different animal models, including T2D models which represent the most common form of diabetes in humans [126], should be encouraged. A potential source of naturally occurring models of diabetes may come from veterinary clinics. Pet dogs, for instance, offer some advantages over laboratory rodents, having size, longevity, and heterogeneous backgrounds closer to humans [127], thus potentially representing an intermediate platform for evaluating novel targets before clinical trials.(2)The general lack of proper glycemic control in animals: while drugs or insulin injections are used to maintain blood glucose within a normal range in humans, this does not usually apply to animals, in which a more explicit severe phenotype is usually preferred.(3)The overall lack of consensus in the protocols for the induction of diabetes in animals, with particular emphasis on dose, age of the animals, duration of the period of observation, and experimental endpoints.

On the clinical side:(1)Few studies provide temporal data on the periods between the onset of diabetes, the onset of DN, and the patient’s age, making any attempt to compare with time windows in preclinical models difficult.(2)Skin biopsies to quantify small fiber density represent a sensitive and objective diagnostic method to detect early signs underlying neuropathic pain in DN patients [128]. Prospective studies showed that the diabetes-induced early degeneration of skin nerve fibers correlated with the duration of the disease [129]. However, the method is still rarely performed in humans, and data on sex differences are not available.(3)Well-established personal, social, and cultural biases associated with gender expectations in experiencing and reporting pain may affect the scoring system in pain questionnaires. This raises the question of the extent to which the sex differences identified in humans genuinely reflect actual neurobiological differences in pain mechanisms or if they are somehow influenced, when not induced, by well-rooted gendered norms in healthcare assessment [116,130].

In general, a major limiting factor in both clinical and preclinical settings concerns pain assessment methods in DN. As shown in Figure 1, proper pain assessment is missing in about 30% of preclinical and clinical studies. In several of these studies, NCV is adopted, both in animals and humans, to assess the severity of DN by detecting large fiber alterations in DN [118]. In clinical settings, NCV analysis has been used to test the efficacy of drugs in counteracting DN and is often combined with tests for loss of vibration perception and mechanical sensitivity [131]. Conversely, the loss of function in the somatosensory system of animals is often unrecognized or not evaluated [132]. When pain thresholds and symptoms are assessed, most of the procedures in animals and humans are hardly comparable. Human pain is usually based on specific questionnaires and Visual Analogue Scales (VAS) that reflect the patient’s subjective perception of pain [133]. Contrariwise, pain in non-communicating diabetic animals is mostly measured by reflexive tests focused on specific pain modalities (i.e., mechanical or thermal sensitivity), which are highly repeatable and relatively simple but are more related to measuring nociception rather than pain perception [134]. To facilitate translational and reverse translational approaches, comparable measurements and assessments in clinical and preclinical studies should be developed. A more comprehensive objective and translatable method in humans is represented by the QST [135]. QST is a set of measurements following mechanical and thermal stimuli of controlled intensity, exploring both small and large fibers and eventually providing a sensory profile of each evaluated subject. Similar approaches have also been validated in animals [136] and recent efforts have been made to develop an appropriate methodology to assess pain sensitivity in different modalities and sexes, overcoming canonical reflexive tests [137].

Finally, besides applying comparable pain assessment methods, more efforts should be made to unify the definition of “controls” across clinical and preclinical studies. Indeed, while the control groups in animal models are always clearly identified as age-matched, sex-matched non-diabetic subjects undergoing the same procedures, the references for patients with DN are highly variable and heterogeneous and may include either non-diabetic individuals or diabetic individuals without DN., Different prevalences in DN within the population of diabetic patients, particularly in cross-sectional studies, may simply reflect the heterogeneity of the reference group. 

## 6. Future Directions

The poor consideration that sex differences in DN, as well as in other diseases, have gained in the past is probably one of the possible reasons for the underlying failures in clinical trials based on findings from animal models. Indeed, in recent decades, different drugs derived from preclinical studies have been tested to counteract painful DN—including sodium channel blockers [138,139], TRPA1 antagonists [140], and antioxidants [141] —with quite variable and/or unsatisfactory results. Future strategies are moving toward the development of tailored gene therapy for painful DN [142], which, together with hormonal approaches based on the protective effects of estrogens [143], may open the route to personalized sex-targeted medicine.

The methodology chosen for the evaluation of DN at both clinical and preclinical level is essential for the proper identification of biologically relevant sex differences and for the subsequent application of this knowledge in translational medicine. A reciprocal exchange between laboratories and the bedside is the necessary, but often missing, step for refining experimental and clinical protocols and for the development of common guidelines for pain assessment in diabetes. Such an approach is required for proper comparisons between models and patients for which the time course of the pathology is quite different and the aspects related to the impact of age across sexes is underestimated [144,145]. More efforts should also be made to define sex differences in the somatosensory system of control animals and “healthy” humans. Do animals and humans display sex differences in NCV? Do they display sex differences in baseline sensory thresholds according to pain modalities? How do stress, environment, physical activity, food intake, and sociality affect sensory behaviors between the two sexes? Do males and females display different responses to analgesic drugs?

While this field is rapidly growing [137,146], a strong understanding of the connections between clinical observations, preclinical data, and the underlying mechanisms is still largely unmet, thus reiterating the need for a more open exchange between models and patients to successfully address sex differences in DN.

## Figures and Tables

**Figure 1 cells-13-02024-f001:**
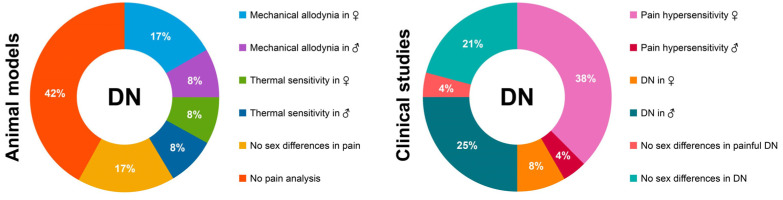
Preclinical to clinical studies in DN and reverse translation—graphic summary of data derived from the analysis of reviewed articles regarding DN at both preclinical and clinical levels.

## Data Availability

No new data were created or analyzed in this study.

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
