# Peer review of "Painful Diabetic Neuropathy: Sex-Specific Mechanisms and Differences from Animal Models to Clinical Outcomes"

_cells, 2024, doi:10.3390/cells13232024_

Round 1
Reviewer 1 Report
Comments and Suggestions for Authors
In this review, the authors address the question of sex-specific differences in painful diabetic neuropathy from a clinic and preclinic point of view. The reporting of the biography and his conclusion/interpretation are easy to follow. The analysis of the literature is of good quality and convincing, conclusions are supported by the observations.
Comment:
Also not completely in line with this revue, since their no evidence of a sex effect (?), the authors may consider in a future review, the putative role of insulin in this context. In some preclinic studies using T1D model or in Human T1D, insulin injections are used to maintain blood glucose within a normal range. This raises the question whether insulin signaling in the CNS is altered by T1D? and vice et versa. Insulin crosses the blood-brain barrier through a transport termed receptor-mediated transcytosis. Insulin receptors are expressed in dorsal and ventral spinal cord neurons where it regulates AMPA induced neuronal damage and modulate AMPA excitatory currents suggesting that insulin may be important to maintain central neuronal function and reduce diabetic neuropathy. Insulin may also control descending serotonin pathway, involve in pain control.
Vital P, Larrieta E, Hiriart M (2006) Sexual dimorphism in insulin sensitivity and susceptibility to develop diabetes in rats. J Endocrinol 190:425–432. https://doi.org/10.1677/joe.1.06596
Spicarova D, Palecek J (2010) Modulation of AMPA excitatory postsynaptic currents in the spinal cord dorsal horn neurons by insulin. Neuroscience 166:305–311. https://doi.org/10.1016/j.neuroscience.2009.12.007
Martin H, Bullich S, Martinat M, et al (2022) Insulin modulates emotional behavior through a serotonin-dependent mechanism. Mol Psychiatry. https://doi.org/10.1038/s41380-022-01812-3
Reviewer 2 Report
Comments and Suggestions for Authors
This article by Merlin et al. provides a comprehensive review of the sex-specific mechanisms involved in painful diabetic neuropathy. The authors highlight key differences in the progression, clinical presentation, and underlying molecular and cellular mechanisms of diabetic neuropathy across sexes. They emphasize the importance of comparing pre-clinical animal models with clinical data to better understand these differences and refine research approaches. The manuscript offers valuable insights into how sex-specific factors could inform the development of more targeted therapies for managing pain in diabetic neuropathy.
Minor points and misspellings
Line 26-17. ‘a number that is expected to increase of about 50% by 2075.’ should be ‘increase by about 50% by 2075.’
Line 27. Is the citation reported correctly?
Line 42. ‘well correlate’. should be "strongly correlate"
Line 47. ‘as well impaired balance’ and 'as' is missing
Lined 65-66. ‘in the same patient along the course of the pathology’
'along' should be modified as ‘during'.
Lined 68-70. ‘The mechanisms bringing to painful or painless DN are still poorly understood and many inflammatory markers did not vary between diabetic patients with and without pain symptoms.’
‘bringing to’ is incorrect. Use ‘leading to.’
‘did not vary’ is overly simplistic. I suggest ‘show no significant variation.’
Lines 83-84. ‘(and actually not only in the pain field)’. No references are given. I suggest to remove the phrase.
Line 92. ‘Biological differences across sexes in the development of chronic pain’. should be simplified as ‘Sex differences in chronic pain development’
Line 94. ‘Sorge and coll.’ should be ‘Sorge et al.’. There are other ‘and coll’ in the test and they all should be ‘et al.’
Line 112. ‘Nevertheless, not everything is different across sexes.’ should be more formal.
Line 261. Remove the comma before [93].
Lines 316-317. ‘Research shows that the male sex may be associated with a higher risk of developing severe diabetic neuropathy with earlier time point [83].’ The phrase ‘with earlier time point’ is unclear.
Line 333. ‘High noise in the literature’ is informal.
Figures 1. is informative; however, they should be uploaded in higher quality. To better interpret the figures, include numerical values next to the labels. Additionally, ensure that the chosen colors are accessible for individuals with color vision deficiencies by following established guidelines for colorblind-friendly design.
Reviewer 3 Report
Comments and Suggestions for Authors
The review study conducted by Merlin et al. describes studies that evaluated sex-specific mechanisms and differences in animal models and clinical trials involving diabetic neuropathic pain. The review covers the most important studies in the literature in a generalized manner, being an important work for studies in the area and for future studies. Some points presented below may be appropriate.
- Na frase “GABAergic tonic currents mediated by α5-GABAA receptors are stronger in sensory neurons of female mice with chronic pain and, unlike male mice, are involved in pro-nociceptive effects”. Is this correct? Is GABA mediating a pro-nociceptive effect? ​​It would be interesting to better explain this opposite effect to the normal in the text.
- The authors do not describe in the manuscript text the search platforms that were used;
- What guidelines were followed for writing the review? Was it the prism? Was the study previously registered in it?
- Na frase “Since PPARγ has been previously linked with female immune system activation, these findings open new avenues in the roles of non-neuronal cells in shaping sensory profiles across sexes in 194 diabetes”. What non-neuronal cells could be involved? It would be important to provide an example of a study;
- Wouldn't a limitation of clinical studies also be the differences in blood glucose levels between participants?
